# Unlocking metabolic insights with mouse genetic diversity

Stewart W C Masson [ID] [1,2,4], Harry B Cutler [ID] [1,2,4] & David E James [ID] [1,2,3] [✉]

**Metabolic diseases are caused by a complex interplay between genetics and the environment, yet many basic studies guiding our understanding of metabolism are confined to single genetic backgrounds, like the C57BL/6 J mouse. Recent studies across different genetic backgrounds have revealed profound phenotypic diversity, questioning the robustness and utility of observations derived from isolated strains. Those embracing genetic diversity will likely discover novel and penetrant mechanisms underlying metabolic dysregulation and disease, and findings may even benefit from increased translatability to humans. The purpose of this commentary is to equip researchers with a practical guide to performing studies across genetically diverse mice, and to highlight some of the important challenges such studies present.**

See also: Metabolomics Methods Commentary Series 2024
See also: W Teale & D Klimmeck

## Introduction

Around the same time that Gregor Mendel's now famous pea experiments were being rediscovered, the first seeds of mammalian genetics were being inadvertently sown (Steensma et al, 2010). Like Mendel, Abbie Lathrop—an esteemed breeder of 'fancy mice' which were considered fashionable pets at the time— kept meticulous records of her crosses such that the parental heritage of each mouse could be traced back to its ancestral strain. At its height, Lathrop's fancy colony housed over 11,000 mice (Steensma et al, 2010) and, in 1902, Lathrop sold mice to Harvard University geneticist William Ernest Castle and his student Clarence Cook Little. The pair went on to conduct the first Mendelian analysis in mammals, and Little eventually formed the Jackson Laboratories. C57BL/6J mice, perhaps the most ubiquitous strain in all of the metabolic research, are direct descendants of Lathrop's 'mouse 57' (Steensma et al, 2010). It seems that much of this history has been lost to the metabolic field, however, since many researchers perform experiments exclusively on the C57BL/6J strain, even though the first standardised mouse experiments took the form of multi-parent, or multi-strain, studies (Little and Tyzzer, 1916; Tyzzer, 1909).

Had we chosen something other than the C57BL/6J mouse as our primary model, the field of metabolism would look very different. A/J mice have shown us that obesity does not always go hand in hand with glucose intolerance (Bachmann et al, 2022; Karimkhanloo et al, 2023), BALB/C mice have revealed the tissue-specific nature of insulin resistance (Nelson et al, 2022), and PWK mice have demonstrated that mice do develop diet-induced NASH akin to humans (Benegiamo et al, 2023). Thus, it is to multi-strain genetic studies, the birthplace of research in mice, that we must return if we are to truly understand the pathogenesis of complex metabolic disease in humans. The use of mouse genetics and associated analytical tools have been superbly reviewed elsewhere (Allayee et al, 2023; Buchner and Nadeau, 2015; Li and Auwerx, 2020; Seldin et al, 2019; Williams and Auwerx, 2015), and so the focus of this article is to instead outline key considerations for metabolism researchers embarking on multi-strain journeys. We discuss how studies capturing metabolic outcomes across genetic diversity can be designed using only a handful of inbred strains and explore how these approaches might be improved in future to optimise the discovery of human translatable disease mechanisms.

## Beyond C57BL/6J

It has been calculated that only a third of the findings from mouse studies can be replicated in humans (Hackam and Redelmeier, 2006). While this finding initially points to inherent biological differences between mice and humans (Perlman, 2016), it also seems plausible that the lack of translation is simply due to a failure of most mouse studies to capture genetic diversity since most metabolic mouse studies are only performed in C57BL/6J mice. While dramatic phenotypic variation between individuals has motivated significant efforts to ensure equal genetic representation in human genomic studies (Bick et al, 2024; Mahajan et al, 2022; Sirugo et al, 2019), similar thinking has been slower to take effect for studies in mice.

Many examples have demonstrated the non-generalisability of C57BL/6J mice in metabolic studies. For example, the entrenched belief that there are no preclinical mouse models of metabolic-associated steatohepatitis (MASH) (Teufel et al, 2016) has recently been challenged by studies identifying several readily available mouse strains that are susceptible to diet-induced fatty liver disease (Benegiamo et al, 2023; Karimkhanloo et al, 2023). When fed a Western diet, wild-derived PWK/PhJ mice developed severe MASH with a degree of fibrosis equivalent to human disease

[1]School of Life and Environmental Sciences, The University of Sydney, Sydney, NSW, Australia. [2]Charles Perkins Centre, The University of Sydney, Sydney, NSW, Australia. [3]School of Medical Sciences, The University of Sydney, Sydney, NSW, Australia. [4]These authors contributed equally: Stewart W C Masson, Harry B Cutler. [✉]E-mail: david.james@sydney.edu.au
https://doi.org/10.1038/s44318-024-00221-2 | Published online: 16 September 2024

 

**Box 1    Web tools and databases**

- Strain selection
  Mouse Phenome Database: https://phenome.jax.org/. Jackson Laboratories curated data across multiple studies. Searchable for phenotypes of interest as well as outlying strains (Bogue et al, 2020).
  Single Nucleotide Polymorphism (SNP) database: https://www.informatics.jax.org/snp. Jackson Lab's database of SNPs across strains. Searchable by gene or genomic region. Useful for identifying strains with SNPs in genes of interest as potential research models where disease-relevant syntenic regions have been identified in human genetic association studies.
- Molecular data integration
  http://diabetes.wisc.edu/. Collection of data generated by the Attie Lab. Includes metabolite, protein and transcript data from plasma, liver, white adipose tissue and whole islet. Useful for selecting which of the Diversity Outbred and Collaborative Cross founder strains to choose.
  https://wren.hms.harvard.edu/DOFounderLivers/. Part of a collection of datasets from the Gygi Lab. Includes male and female liver proteomics from the 8 DO founder strains. Useful for selecting strains for liver-based metabolic investigations.
- Systems genetics
  https://systems-genetics.org/. Wide-ranging BXD and Collaborative Cross (CC) mouse population resource. Includes genetic associations for phenotypic, transcriptomic and longevity data for BXD strains, as well as transcriptome, proteome and phenome data for CC founder strains in different experimental contexts (Li et al, 2018).
  https://qtlviewer.jax.org/. A web tool for QTL analysis of publicly available collaborative cross and diversity outbred datasets. Includes a collection of physiological, proteomic and transcriptomic data from various tissues. Also includes estimates of genetic contributions from the DO and CC founder strains (Vincent et al, 2022).
- Exploratory searches
  https://genenetwork.org/. Extensive collection of genes, molecules, and higher order gene function and phenotypes from humans, mice, rats, flies, and plant species (barley and Arabidopsis). Integrated with SNP and QTL datasets to allow web-based mapping of each trait (Mulligan et al, 2017).

pathology, and exhibit considerable transcriptomic overlap with human liver disease. While studies like this highlight marked differences between mouse subspecies and wild-derived strains compared to their traditional laboratory counterparts, one need not climb so far on the phylogenetic tree to find striking phenotypic variation (Simon et al, 2013). The C57BL/6N sub-strain varies from its C57BL/6J cousin by only 51 coding variants, however, it exhibits lower energy expenditure and enhanced insulin secretion even under identical chow-fed conditions (Simon et al, 2013). Taking C57BL/6J as the representative for all mice, when even some of its closest relatives exhibit divergent metabolic phenotypes, is problematic. Strikingly, even the effects of genetic polymorphisms differ by strain (Sittig et al, 2016), highlighting the overwhelming influence of genetic background on metabolism. Thus, widening research horizons to encompass a broader landscape of mouse genetic diversity is quintessential to the pursuit of robust mechanistic discoveries.

## Choosing the right strain/s

Largescale genetic and phenotypic analysis of metabolic traits in mice have been popularised through the development of large inbred panels like the BXD (Roy et al, 2021; Williams et al, 2016), UM-HET3 (Bou Sleiman et al, 2022), Hybrid Mouse Diversity Panel (HMDP) (Parker et al, 2019; Parks et al, 2015) and Collaborative Cross (CC) (Aylor et al, 2011). These have been excellently reviewed in detail (Ashbrook et al, 2021; Buchner and Nadeau, 2015; Churchill et al, 2004). More recently, Diversity Outbred (DO) mice (Chesler et al, 2016; Churchill et al, 2012; Masson et al, 2023) have featured in many analyses of the genetic architectures of complex traits. Analyses using these resources require large numbers of mice to achieve sufficient power for genetic analyses, however, many insightful multi-strain studies can in fact be accomplished using more targeted approaches. Importantly, this hinges on the public release of data to facilitate the selection of the most appropriate strains for a given analysis. Fortunately, a number of online resources already exist to assist metabolism researchers new to the field of mouse genetics (Box 1). As discussed above, great progress has been made in the MAFLD/MASH field using such approaches, and with further effort, one can envisage similar breakthroughs in the discovery of mouse models that replicate other metabolic traits. Indeed, Morahan and colleagues recently identified a collaborative cross-strain that recapitulates Diabetic Retinopathy, something not observed in most other strains (Weerasekera et al, 2015). Resulting mechanistic studies will benefit greatly once susceptible strains have been identified, as it will be possible to compare responses in disease-prone versus disease-refractory animals.

To simplify the process of strain selection, here we have compiled a collection of inbred mouse strains with diverse metabolic responses to either high-fat or Western diets that could be used to investigate specific metabolic phenotypes (Fig. 1). For example, comparing DBA/2 J and BXH9/TyJ mice might provide insight into mechanisms underpinning systemic glucose homoeostasis. While both strains develop obesity, fatty livers and hyperinsulinemia following a high-fat diet, the DBA/2J is largely protected from glucose intolerance (Montgomery et al, 2013). Such studies, therefore, prove a resource-effective approach with which to delineate individual components of the metabolic syndrome.

Investigations in A/J mice provide clear examples of this, where the effects of diet-induced obesity have been separated from more severe metabolic dysfunction, exemplified by the work of Joseph Nadeau (Burrage et al, 2010; Hoit et al, 2002; Singer et al, 2004) and Richard Surwit (Collins et al, 1997; Seldin et al, 1994; Surwit et al, 1995). Many phenotypic differences between strains are well established and so studies seeking to elucidate molecular drivers of these differences need only cast an 'omics lens over these strains to identify potential mechanisms. Considering that some strains demonstrate inherent phenotypic differences at baseline while others display altered susceptibilities to

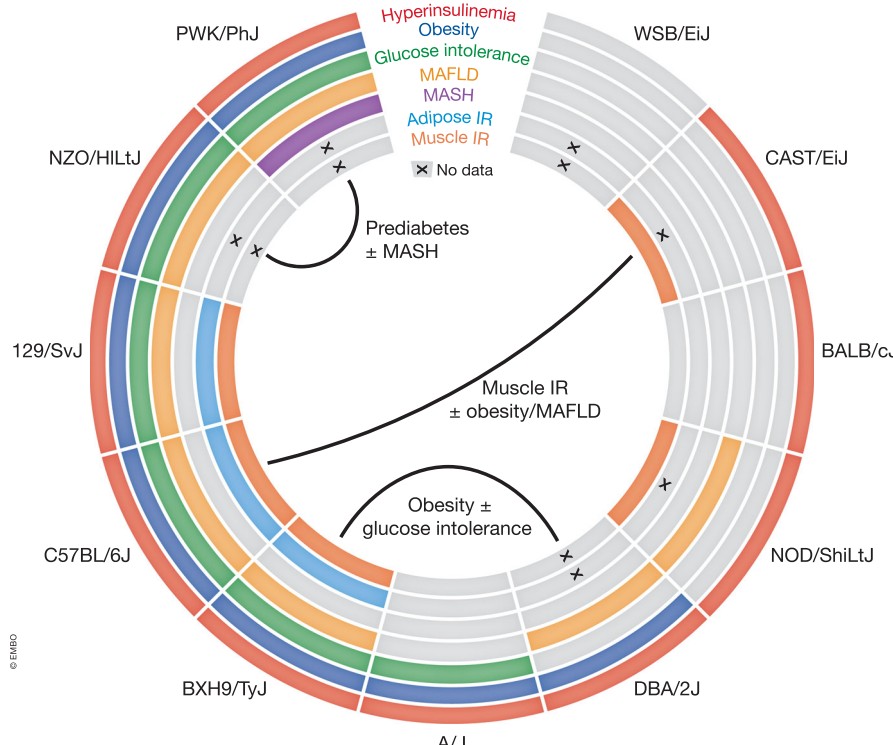

**Figure 1. Exciting studies with genetic diversity can be designed using as few as two strains.**

Susceptibilities to metabolic defects induced by Western or High Fat diets are shown for 11 strains commonly used in the metabolic literature (Bachmann et al, 2022; Benegiamo et al, 2023; Karimkhanloo et al, 2023; Nelson et al, 2022; van Gerwen et al, 2024). Lines connect several example strains for which interesting comparisons could be made, although many others are possible. Annotations denote possible comparisons. Readers are encouraged to review the literature for other valid strain comparisons.

intervention-induced differences, the possibilities to identify novel molecular drivers of disease are boundless.

The founder strains of the collaborative cross and diversity outbred populations have been extensively phenotyped and, given the ability to replicate genetic background across studies, offer a valuable publicly available resource for the identification of relevant mouse strains. Moreover, traits in these strains can be bioinformatically linked to the more complex DO and CC populations thanks to their shared genetics (Churchill et al, 2012). Databases containing intermediate phenotypes, including transcript and proteomic data of certain tissues, mean that any of these 8 strains, as well as those from other diverse mouse panels, make ideal additions to a given study (see Box 1). Utilising these strains and resources, therefore, has advantages for both the user and the community through the public release of new data that will inevitably improve the design and outcomes of future experiments.

## Practical considerations

### Experimental design and data interpretation

The phenomenal metabolic diversity between different mouse strains poses several challenges to experimental design and resulting data interpretation. For example, exercise capacity varies significantly between different inbred mouse strains (Courtney and Massett, 2012; Moore et al, 2023; Williams et al, 2016). Studies examining the effects of genetics on exercise metabolism must be careful, therefore, to assess the exercise performance of different strains relative to their respective maximum capacities (e.g. 80% of max running speed) as this may otherwise lead to a misinterpretation of results. One may conclude that strains exhibit differential exercise-induced signalling because of genetic differences in kinase activity, whereas in reality, each strain was subjected to different *amounts*

of relative exercise. This phenomenon is well appreciated in humans where interpersonal differences in $VO_2$ max are frequently considered (Blazev et al, 2022; Broome et al, 2022), and may seem common-sense. However, considerations such as this as critically important to ensuring that experiments generate robust comparisons between strains. Almost all considerations depend on the biological question of interest, and investigators must be vigilant to select the most optimal experimental design to avoid challenges with downstream data interpretation.

Multi-strain studies generally collect more data than can be easily analysed, and when faced with numerous unexpected phenotypic correlations, it can be exceedingly difficult to piece the biological puzzle together. The fact that the findings of such studies frequently challenge ingrained dogmas will inevitably lead to short-term difficulties as the nuances of genetic diversity are embraced. However, it is imperative that the metabolic community embrace the complexities of genetic diversity to gain novel insight into the biological processes underlying phenotypic changes.

One classic example where multi-strain studies have exposed the true complexity of a biological process relates to the belief that calorie restriction (CR) has universally beneficial effects on extending lifespan (Liao et al, 2011). Coincidentally, this was the study that first motivated our lab to venture into multi-strain studies. By investigating the effects of CR beyond the C57BL/6J background, Liao et al showed that while some inbred mouse strains exhibit lifespan extension with CR, many actually exhibited increased mortality. The great potential of this finding to enhance understanding of the mechanism by which CR extends lifespan, however, was overshadowed by criticisms relating to poor statistical power and inappropriate study design (Mattson, 2010; Swindell, 2012). Importantly, the study did not include the C57BL/6J strain as a control and so the potential detrimental effects of CR were explained away as having resulted from too severe an intervention. Although legitimate, such concerns can easily be prevented by including a well-studied strain as a control, enabling the biological importance of study findings to be fully appreciated. Finally, it is essential that experiments be designed with statistical power in mind so that meaningful comparisons can be made. This should be

rigorously assessed at the level of funding decisions, as study power is critically important to the feasibility of studies to elucidate new biology.

## Multi-strain 'omics analysis

Another important consideration relates to the challenges of performing multi-omics studies across genetically diverse mice. Metabolism research has benefited significantly from developments in nucleotide sequencing and mass-spectrometry-based methods and, together with mouse genetic diversity, this has led to the emergence of systems genetics. These have been reviewed here (Allayee et al, 2023; Buchner and Nadeau, 2015; Li and Auwerx, 2020; Seldin et al, 2019; Williams and Auwerx, 2015; Williams et al, 2016) and, for an excellent overview of how proteomics specifically has contributed to these efforts, see (Molendijk and Parker, 2021). Molecular traits like protein expression are born from a dynamic integration of genetics with the environment and, as such, offer highly appealing insight into the mechanisms underpinning defective metabolism. Here we provide recommendations for investigators interested in applying such technologies to conduct systems genetics in mice.

While the following considerations are specifically relevant to mass-spectrometry-based omics, they should apply equally to other omics workflows. Firstly, it is important to randomise the processing and analysis of samples from various strains and diets to avoid issues with collinearity. If sample numbers exceed that which can be prepared in a single batch, include a common set of 'sample preparation controls' that can be compared between batches. Second, it is advisable to run a single pooled control sample periodically throughout the entire run (once every 20 samples) to account for potential drift in instrument performance (Kim et al, 2021). Finally, prior to data analysis, it is essential to run quality control checks on the data, including assessment of batch effects using multi-dimensional scaling approaches such as principal component analysis (PCA) or uniform manifold approximation and projection (UMAP). If batch effects are present, the implementation of correction tools to minimise their effects on interpretation should be considered (Haghverdi et al, 2018; Kim et al, 2021). In some cases where batch effects are too significant to be adequately corrected, it may be necessary to consider re-running samples with appropriate controls.

Genetic variation across strains can also cause issues depending on the experimental approach. For example, multiplexing approaches like tandem-mass tag (TMT) labelling can reduce sample numbers for MS analysis but caution needs to be exercised in analysing data from genetically diverse mice where coding variants may give rise to changes in the peptide sequence. Similar issues may also arise in aligning DNA or RNA reads to a reference when polymorphisms exist. Most reference libraries are generated using the C57BL/6J mouse genome, and for TMT-based peptide quantification, this has the potential to confound data analysis as coding variants may lead to the resulting peptide being quantified as zero rather than not detected. If not accounted for, this may lead to the incorrect interpretation that strains with mutated peptide sequences have decreased protein abundance. Fortunately, this can be avoided by searching spectra against libraries which exclude polymorphic peptide sequences (Keele et al, 2021; Xiao et al, 2022). This issue is not present when using label-free quantification as mutated peptides will not affect protein quantification, provided that other peptides with which to quantify the protein have been detected. Similar precautions should be considered across other "omics" technologies. For example, when performing sequencing (e.g. RNA-seq or ATAC-seq) from multiple organs in large cohorts, thoughtful approaches to library preparation and pooled sequencing strategies can improve data quality in downstream analyses.

### Challenges of working across strains

Our laboratory only embraced genetic diversity to study metabolism relatively recently, having previously worked exclusively with C57BL/6 J mice. Many studies in our lab are now performed in multiple inbred strains and even an outbred population. Box 2 contains a selection of pointers

---

**Box 2  Mouse handling in multi-strain studies**

- Pilot experiments across study strains. We and others encountered many inter-strain differences during the assay development of an in vivo method for measuring insulin action in individual mice (Cutler et al, 2024). For example, the required drug dose to ensure adequate depth and duration of anaesthesia. NOD/ShiLtJ are extremely sensitive to isoflurane, and their heart rate will drop rapidly if not carefully dosed. Others have reported similar strain-dependent variations in ex vivo lipolysis assays (Collins et al, 1997).
- Scale assays for robustness across many mice, as well as financial constraints. In our experience, the most worthwhile optimisations relate to performing assays in 96-well plates and evaluating how the impact of the increased time required to perform assays over large sample sizes may influence results.
- Wild-derived strains (e.g. WSB/EiJ and CAST/EiJ) are highly active and prone to fighting injuries. Include extra enrichment in the form of bedding, cardboard tubes and wooden blocks to reduce aggression between these mice.
- WSB/EiJ mice, in particular, are extremely active and can quickly derail weekly weighing or a glucose tolerance test if not managed correctly. Place WSB cages in large plastic tubs prior to opening to prevent them from escaping onto the floor and use cardboard tubes with one closed (but ventilated) end to hold the mice if collecting blood samples is necessary.
- Strains which develop extreme obesity, such as the NZO/HlLtJ, may become unable to groom effectively and will need extra care, such as removing build-up in and around their prepuce. Investigators using NZO mice should also be aware that they may need to be culled earlier than other strains when performing long-term diet studies, given significant adverse metabolic phenotypes (Bachmann et al, 2022).
- Large studies including multiple strains will require many researchers. There is evidence that different animal handlers can influence physiological outcomes in mice (Sorge et al, 2014). Where possible we recommend limiting the number of handlers per study and ensure equivalent technical skill between team members.
- Female mice are prolific groomers and often have very little fur or whiskers. Although adverse effects are rare, this is something that should be accounted for in any ethics application since standard mouse grooming criteria will result in unnecessary mouse euthanasia.
- Different strains of mice (particularly females) pull soft food down from the hopper and incorporate it into the bedding. To avoid wastage, give less food more frequently and change bedding regularly to avoid skin irritation.

---

curated by our animal team from personal experience and across the literature, which we believe will be helpful for researchers eager to embark on their own multi-strain adventures.

## Beyond genetic mapping

An ever-growing tool kit of genetically diverse mouse populations has enabled high-resolution genetic mapping of complex traits and molecular phenotypes, such as protein and metabolite abundance (Al-Barghouthi et al, 2021; Chella Krishnan et al, 2023; Keller et al, 2018; Keller et al, 2019; Linke et al, 2020; Masson et al, 2023; Price et al, 2023a; Price et al, 2023b; Takemon et al, 2021; Xiao et al, 2022; Zhang et al, 2023). However, genetic mapping is only one of many powerful uses for these populations. We direct interested readers to the following reviews to learn more about the development and genetic architecture of the most recently developed mouse genetic resource, the Diversity Outbred (DO) population (Chesler et al, 2016; Churchill et al, 2012; Gatti et al, 2014). Several other populations, including the BXD, HDMP (Buchner and Nadeau, 2015) and Swiss Outbred (Chia et al, 2005) resources, contributed greatly to the fundamental characterisation of complex traits through approaches such as phenome-wide association studies.

A prominent feature of genetically diverse mouse panels is their capacity to reveal penetrant, genetically determined disease mechanisms. Power calculations suggest sample sizes an order of magnitude lower than those required in human studies (Gatti et al, 2014), largely due to heightened control over environmental variation. In addition, their phenotypic variation makes them ideal for phenotypic discovery and preclinical testing of potential therapeutics or interventions. It has even been suggested that diverse mouse populations should become a routine part of the clinical trials pipeline (Nadeau and Auwerx, 2019). Reframing of DO studies in this way will hopefully encourage increased uptake amongst labs that lack the capacity to conduct large (>400 mice) complex trait genetic mapping studies. For US-based researchers, DO mice can be sourced relatively easily from Jackson Laboratories (https://www.jax.org/strain/009376), and Jackson Labs have previously supported pilot programmes for interested researchers

(https://www.jax.org/jax-mice-and-services/diversity-outbred-grant).

## Validation bottlenecks

One of the most common reasons for paper rejection is 'lack of mechanism'. This is, in fact, one of the greatest strengths of the mouse because it provides enormous opportunity for mechanistic analyses. However, with the expansion in multi-omics technologies, we are now facing a validation bottleneck; many investigators who have used 'big data' approaches to generate interesting hypotheses struggle to apply current validation tools to genetically diverse mouse populations. Although many validation approaches exist, including knockout and knock-in mouse models, adeno-associated viruses (AAVs), lipid nanoparticles (LNPs) and anti-sense oligonucleotides (ASOs), these have been predominantly optimised using C57BL6/J mice, and it is unclear how simple it will be to apply such tools to other strains.

While knockout mice are a workhorse of metabolic research, results from such experiments are often not physiologically relevant in the context of natural genetic variation in protein abundance. Additionally, the time constraints of generating knockout mice in different strains or performing other gain/loss of function studies is not compatible with the high throughput nature of target generation that has been facilitated by systems genetics. In our view, the metabolism space has just as much to benefit from reinvigorated experimental approaches as neuroscience has from the development of opto- and chemo-genetic tools (Campbell and Marchant, 2018; Rost et al, 2017). Here we discuss future needs to perform validation across multiple strains, although even studies in C57BL/6 J mice stand to benefit from the development of enhanced validation approaches.

Current in vivo metabolic validation approaches such as knockout or transgenic models are largely limited to C57BL/6J mice, a reality that is especially restricting given the potential utility of alternate mouse strains. It is difficult to envisage an experiment in standard-issue C57BL/6J mice to alleviate MASH when they are not susceptible to this pathology. Current approaches also overlook the potential advantages posed by performing experiments in different strains. For example, some strains may be protected from a particular metabolic defect because they overexpress a particular

protein, while others may be susceptible because they are low expressers. The most powerful validation of this would be to knock-down expression in the protected strain in a tissue-specific manner and over-express in the latter, however, such experiments are challenging to perform at present.

This issue is further compounded in scenarios where researchers wish to perturb gene expression in a specific tissue, since tissue-specific Cre lines exist almost entirely on the C57BL/6J background and would require investigators to perform experiments in a mixed background or perform time-consuming backcrossing experiments. Although a solution to these challenges is not yet in the mainstream, recent developments in the adeno-associated virus (AAV) and lipid nanoparticle (LNP) fields may greatly assist metabolism researchers. AAVs are inert, non-replicable viral vectors that can be used to overexpress or silence specific genes in specific tissues. Although AAVs have traditionally been used to target the liver, recent advances in designing tissue-specific promoters and cis-regulatory elements (Huang et al, 2017; Sarcar et al, 2019) provide hope that it will be possible to perturb gene expression in other metabolic organs. Fascinatingly, machine learning methods have recently been used to develop potent and tissue-specific promoters in *Drosophila* (de Almeida et al, 2024), and similar studies in mice would likely be highly beneficial. Like AAVs, LNPs offer an approach to deliver genetic material to tissues, although are reportedly less immunogenic (Cheng et al, 2020). Although tissue-specificity is less well-established for LNPs compared to AAVs, a number of studies have been successful in directing particles to target organs by modifying specific lipids contained within the LNPs (Xu and Xia, 2023). Further work is required to determine how effectively AAV and LNP approaches will work across different genetic backgrounds, as unpublished data from our lab indicates that the efficiency of AAV transduction varies by mouse strain. Regardless, the development of more effective methods for manipulating gene expression in vivo is essential if we are to reduce the barriers to experimental validation.

## Conclusion

There is a growing view that data obtained from mouse studies does not translate to humans and it has even been suggested that funding agencies should cease funding mouse experiments at the expense of human studies. Here we make the case, as has been done by

---

**Box 3    Take home messages**

- C57BL/6J are not representative of all mice, the same way that one human does not capture the metabolic diversity found across populations. There is much to be learnt by studying different strains.
- Selecting the right strain for your research question can be difficult but there is a growing collection of databases to help. Even comparing two strains can provide insight into metabolism.
- Assays should be piloted and optimised across strains to ensure fair comparisons.

- There is more to Diversity Outbred mice than genetic mapping. We believe they hold great potential as an intermediate step between preclinical mechanistic studies and human clinical trials.
- The tools required to validate systems genetic discoveries or test hypotheses across strains are lacking, and further investment is needed.

---

others referenced throughout this commentary, that this 'poor translatability' is largely due to the use of one single mouse strain in the majority of metabolic science and, for this reason, the mouse has been falsely accused. By expanding mouse research into different genetic backgrounds, not only will the research become more translatable, but it will also stimulate new paradigms and molecular discoveries that could enrich our understanding of human biology immeasurably. However, expanding mouse research beyond the C57BL/6J mouse is not without challenges. Here we provided a number of foundational recommendations ranging from strain selection and experimental design to data interpretation and avenues for future research (Box 3). We hope that this provides a practical guide for researchers and inspires the development of new methods required to unlock the full potential of mouse genetics.

## Peer review information

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

## Acknowledgements

We would like to thank the always-helpful Marcus Seldin and Søren Madsen for their insight while writing this commentary. We would also like to thank Kristen C Cooke, Meg Potter and Niamh R Craw for their input regarding handling considerations for mice of different genetic backgrounds. This work was supported by an Australian Research Council Laureate Fellowship (to DEJ), and an Australian Government Research Training Program Scholarship (to HBC). The content is solely the responsibility of the authors and does not necessarily represent the official views of the Australian Research Council.

## Author contributions

**Stewart W C Masson**: Conceptualisation; Visualisation; Writing—original draft; Project administration; Writing—review and editing. **Harry B Cutler**: Conceptualisation; Visualisation; Writing —original draft; Project administration; Writing— review and editing. **David E James**: Conceptualisation; Supervision; Visualisation; Writing—original draft; Project administration; Writing—review and editing.

## Disclosure and competing interests statement

The authors declare no competing interests.

