## [Peer Review File · The EMBO Journal]

Unlocking Metabolic Insights with Mouse Genetic Diversity

Stewart Masson, Harry Cutler, and David James

Corresponding author: David James (david.james@sydney.edu.au)

Review Timeline:

Submission Date:	12th Jun 24
Editorial Decision:	15th Jul 24
Revision Received:	23rd Jul 24
Editorial Decision:	30th Jul 24
Revision Received:	30th Jul 24
Accepted:	1st Aug 24

Editor: Daniel Klimmeck

Transaction Report:

Dear Stewart, dear David, dear Harry,

Thank you again for sending us your commentary article for the metabolism advice series. As mentioned, we have asked two dedicated experts to assess your manuscript, and in the meantime got feedback from both of them, which I enclose below FYI.

As you will see, the experts much appreciate the perspective piece and find it timely and worth publishing. They also provide constructive feedback on how to further improve it by advancing the discussion by adding specific examples, making the text more concise, as well as clarifying a number of aspects, including indicated references.

I hope you will find the comments helpful. I am sure that an amended version incorporating the suggestions made by the referees will be highly noted and appreciated. I would thus like to invite you to submit such a revised version using the link enclosed below.

Please let me know in case I can be of any help with this.

with
Best wishes,

Daniel

Daniel Klimmeck, PhD
Senior Editor
The EMBO Journal

Referee #1:

In this commentary, Masson et al. used previous mouse studies to illustrate the limitations of experiments conducted on animals with a single genetic background. Since phenotypic traits varied across strains with different genetic backgrounds, they advocate that using genetically diverse animals can help explore novel mechanisms underlying phenotypes of interest and improve the translatability of research findings in animal models to humans. By reviewing existing studies, they concluded that susceptibility to metabolic defects induced by Western or high-fat diets varies across 11 mouse strains, and provided suggestions for selecting suitable mouse strains to investigate different metabolic disorders.

This is an important piece of work, because many (>90 %) of the biomedical researchers ignore the fact that mouse genetics has an important impact on the outcome of their studies. Performing experiments in a single genetic background is one of the reasons of poor generalizability of data. Therefore, this work should be published, even perhaps as a review, instead as a commentary.

Major comments:

1. It is important to distinguish between a Western diet and a high-fat diet, as they are not the same. Different studies also use different diets, so it needs to be verified that the correct diet is indicated when describing previous work. For example, in line 72, the food they used is a Western diet instead of a high-fat diet, and in line 110, the study uses a high-fat diet.
2. in line 155, the website <https://systems-genetics.org/> includes transcriptome, proteome, and phenome data of Collaborative Cross (CC) founders with high-fat diet and Western diet challenges.
3. Provide a citation for each webtool listed in Box1
4. Line 134: there is no reference (or examples) to existing Databases. It might also be worth to cite with examples existing resources for genotypic data: strain variants, genomic annotations and so on as those resources can be very useful in the analyses framework that those genetically diverse mice can be used for.
5. Line 196-214: This section is a bit vague and doesn't provide real indications for what is suggested for experimental design

and results interpretation. The generation of a multitude of results should be presented with an increased emphasis on the associated increased potential in understanding biological processes underlying phenotypic changes. Now there are several very general sentences that don't provide added value to the review.

6. Lines 226-229: This long explanation about some more trivial maths could eventually be rephrased by indicating for example some real case-study numbers and how to get to those numbers. This would fit to the framework of this work, where one of the goals is to provide indications on how to best conduct mouse studies. One could provide examples of power calculation needed to estimate the required number of samples and strains. It would also be very interesting to have a comparison with human populations, to point out that the controlled experimental settings in mice allow for much lower sample size than humans to still reach adequate statistical power.

7. Line 233: Instead of focusing exclusively on how to prepare a set of "sample preparation controls", it may be probably even more useful to provide reference or recommendations on how to ideally split strain samples across analysis, to be able to account for batch effects and avoid collinearity (instead of just mentioning randomization). Since this section is about batch effects, it would be useful to reference tools that can be used to address potential batch effect at the data analysis step.

8. I agree with the authors that the DO population is a valuable resource for mouse genetic studies. However, other mouse populations, such as CCs, BXD, UM-HET3, and HMDP, are also important. These mouse populations can also bridge the gap between studies using the C57BL/6J strain. Therefore, it should not be concluded that the DO population is the best one, it is one of the useful ones. Additionally, reference 12 does not make a statement that the DO population should be a routine part of the clinical trials pipeline.

9. Line 301: This statement should be toned down, as other populations contributed to fundamental characterizations of complex traits: BXDs, HMDP, Swiss Outbreds, CCs, etc ...

10. Lines 318-319: It is not clear why the authors choose those specific examples. Also, in this section a mention to potential genetic difference between mouse and humans for the targets to test is missing. This might not necessarily represent a problem but it can be something to consider in particular for some genes. For example, there are genes identified as good targets in mouse with no orthologs, or with more than one ortholog in humans.

11. Lines 331-333: As stated now, it looks like tools to experimentally validate hypothesis in the field of metabolism are missing although many options exist. A reference supporting this statement is required or the authors should expand more with a set of specific examples compared to what exist for example in the field of neurosciences as reported in the text.

12. Line 347: This statement is needs to be reformulated. Widely used techniques exist to perform LOF/GOF studies or introduce specific mutations in strains with different genetic backgrounds exist.

Minor comments:

1. Line 43-44: it could be useful to cite some papers contrasting the examples cited in lines 40-42 but in other strains other than C57BL/6J.

2. Line 73: it might be worth to also point out the very large differences naturally existing between mouse subspecies and with more wild-derived strains, such as CAST as also pretty clear from Fig. 1.

3. Line 80: only mentioning genetic analyses as justification for large populations might be reductive as this also concerns for example association studies between molecular traits and phenotypes (such as ePheWAS) and so on.

4. Line 85-86, more representative papers after each mouse population should be added.

5. Line 93: As a minor comment, as the transition is nor formulated from the sentence before it implies that we are easily able to just identify those particular strains. But this assumes we have access to prior data from a set of strains that can vary in size according to the needs. This is partially addressed later (line 129) by mentioning that some large databases of phenotypes exist.

6. Line 139: it might not be clear for some readers what is the benefit for the community if it is not specified that this comes through the public release of the generated data.

7. Line 155: this resource include also studies conducted on other set of strains other than the BXDs.

8. Line 230: this applies in particular to MS even though this section also concerns DNA-sequencing (including RNA-seq if understood correctly). Also, since samples preparation is chronologically before data generation, we recommend to invert the order.

9. Line 216, should "-omics" be "multi-omics"?

10. Line 221, <https://www.science.org/doi/10.1126/science.aad0189>, this paper is also an example for proteomic study in multi-strains

11. Line 242: As a minor comment, for non-MS expert it might be useful to explain what being qualified as zero means.

12. Line 271, missing reference

13. Line 317: As a minor comment, this is repeated from Line 309

Referee #2:

Summary

The commentary by Masson et al. covers the topic and importance of genetic diversity in experimental mouse models in relation to metabolic phenotypes. The authors initially review the position of the C57Bl/6J mouse line as the still most prominent rodent model in metabolism research, highlighting problems in the generalization of C57Bl/6J data to other strains. Then, an overview over other existing mouse strains is provided, emphasizing their respective metabolic phenotypes and potential areas of application. Furthermore, practical considerations are mentioned that may influence the choice of particular mouse strains and the impact on data interpretation. The authors argue for multi strain "omics" analysis to broaden the data basis and increase the translational impact of obtained mouse studies for human disease conditions. Finally, the authors propose the more extensive

use of Diversity Outbred (DO) mice in metabolic studies as their phenotypic variation may be ideal as a preclinical test environment for new therapeutic compounds. Overall, the authors aim to provide researchers with a practical guide when it comes the choice of the best possible mouse model for metabolic research questions and to discuss potential bottlenecks in this context.

General comments

Given the absolute requirement for translatability of mouse data into human relevance in drug development, the choice of proper model systems remains a critical issue, particularly in the field of complex diseases including obesity, diabetes and related metabolic dysfunction. In this respect, the authors address an interesting and relevant topic in biomedical research. The manuscript is well-structured, concise, and easy-to-read. The figure provides a first and helpful guidance to potentially relevant mouse models in this field. As some aspects in the second half of the manuscript ("challenges of working across strains", "validation bottlenecks") appear somewhat trivial in some parts, the manuscript would strongly benefit from the inclusion of more concrete and successful examples of DO animals in translational research ("In which case did the use of DO animals really make a difference in drug development?"). The addition of specific examples will underline the author's main arguments and make a more convincing case in general.

We thank the reviewers for their thorough evaluation of the manuscript. Each comment is copied below and our responses are in blue. We have amended the manuscript in each case.

Referee #1:

Major comments:

1. It is important to distinguish between a Western diet and a high-fat diet, as they are not the same. Different studies also use different diets, so it needs to be verified that the correct diet is indicated when describing previous work. For example, in line 72, the food they used is a Western diet instead of a high-fat diet, and in line 110, the study uses a high-fat diet.

Thank you for pointing this out. We have checked all references where a specific diet is mentioned and ensured that this is accurately defined throughout. Corrections for the specific articles you mention have been made on lines 67-70 and 107-109.

2. in line 155, the website <https://systems-genetics.org/> includes transcriptome, proteome, and phenome data of Collaborative Cross (CC) founders with high-fat diet and Western diet challenges.

Data relating to CC founder strains is now referenced for this resource in Box 1. This now reads:

- Wide ranging BXD and Collaborative Cross (CC) mouse population resource. Includes genetic associations for phenotypic and, transcriptomic data, and longevity data from the National Institute on Aging Interventions Testing Program among others for BXD strains, as well as transcriptome, proteome and phenome data for CC founder strains in different experimental contexts (42).

3. Provide a citation for each webtool listed in Box1.

Citations are now provided for all resources for which a published manuscript could be located, following the description of each resource in Box 1. This includes: <https://phenome.jax.org/>, <https://systems-genetics.org/>, <https://qtlviewer.jax.org/> and <https://genenetwork.org/>.

4. Line 134: there is no reference (or examples) to existing Databases. It might also be worth to cite with examples existing resources for genotypic data: strain variants, genomic annotations and so on as those resources can be very useful in the analyses framework that those genetically diverse mice can be used for.

The JAX informatics resource (<https://www.informatics.jax.org/snp>) has been included within Box 1 to enable investigators to query strain-specific genetic polymorphisms. This entry in Box 1 reads:

- Single Nucleotide Polymorphism (SNP) database: <https://www.informatics.jax.org/snp>. Jackson Lab's database of SNPs across strains. Searchable by gene or genomic region. Useful for identifying strains with SNPs in genes of interest as potential research models where disease-

relevant syntenic regions have been identified in human genetic association studies.

5. Line 196-214: This section is a bit vague and doesn't provide real indications for what is suggested for experimental design and results interpretation. The generation of a multitude of results should be presented with an increased emphasis on the associated increased potential in understanding biological processes underlying phenotypic changes. Now there are several very general sentences that don't provide added value to the review.

One of the greatest challenges of systems biology relates to how experiments can be designed to bridge the gap between largescale data collection and biological mechanism. While the generation of a multitude of results can in theory lead to greater understanding, it can be incredibly challenging to analyse these data in an unbiased manner. We have amended the text to more precisely address the above concern by focussing on advice relating to statistical power and the inclusion of robust controls. The text now reads:

- Multi strain studies generally collect more data than can be easily analysed, and when faced with numerous unexpected phenotypic correlations, it can be exceedingly difficult to piece the biological puzzle together. The fact that the findings of such studies frequently challenge ingrained dogmas will inevitably lead to short-term difficulties as the nuances of genetic diversity are embraced. However, it is imperative that the metabolic community embrace the complexities of genetic diversity to gain novel insight into the biological processes underlying phenotypic changes.

One classic example where multi strain studies have exposed the true complexity of a biological process relates to the belief that calorie restriction (CR) has universally beneficial effects on extending lifespan (49). Coincidentally, this was the study that first motivated our lab to venture into multi strain studies. By investigating the effects of CR beyond the C57BL/6J background, Liao et al showed that while some inbred mouse strains exhibit lifespan extension with CR, many actually exhibited increased mortality. The great potential of this finding to enhance understanding of the mechanism by which CR extends lifespan, however, was overshadowed by criticisms relating to poor statistical power and inappropriate study design (50, 51). Importantly, the study did not include the C57BL/6J strain as a control and so the potential detrimental effects of CR were explained away as having resulted from too severe an intervention. Although legitimate, such concerns can easily be prevented by including a well-studied strain as a control, enabling the biological importance of study findings to be fully appreciated. Finally, it is essential that experiments be designed with statistical power in mind so that meaningful comparisons can be made. This should be rigorously assessed at the level of funding decisions, as study power is critically important to the feasibility of studies to elucidate new biology.

6. Lines 226-229: This long explanation about some more trivial maths could eventually be rephrased by indicating for example some real case-study numbers and how to get to those numbers. This would fit to the framework of this work, where

one of the goals is to provide indications on how to best conduct mouse studies. One could provide examples of power calculation needed to estimate the required number of samples and strains. It would also be very interesting to have a comparison with human populations, to point out that the controlled experimental settings in mice allow for much lower sample size than humans to still reach adequate statistical power.

We have added a sentence to emphasise the enhanced power of studies in mice compared to humans. This reads:

- A prominent feature of genetically diverse mouse panels is their capacity to reveal penetrant, genetically determined disease mechanisms. Power calculations suggest sample sizes an order of magnitude lower than those required in human studies (68), largely due to heightened control over environmental variation. In addition, their phenotypic variation makes them ideal for phenotypic discovery and preclinical testing of potential therapeutics or interventions.

7. Line 233: Instead of focusing exclusively on how to prepare a set of "sample preparation controls", it may be probably even more useful to provide reference or recommendations on how to ideally split strain samples across analysis, to be able to account for batch effects and avoid collinearity (instead of just mentioning randomization). Since this section is about batch effects, it would be useful to reference tools that can be used to address potential batch effect at the data analysis step.

Discussion of the method of randomisation has been made more specific, and we have included specific discussion relating to assessing and correcting for batch effects. We have also broadened the discussion to hopefully encompass other (non-mass spectrometry-based omics approaches). The text now reads:

- While the following considerations are specifically relevant to mass spectrometry-based omics, they should apply equally to other omics workflows. Firstly, it is important to randomise processing and analysis of samples from various strains and diets to avoid issues with collinearity. If sample numbers exceed that which can be prepared in a single batch, include a common set of 'sample preparation controls' that can be compared between batches. Second, it is advisable to run a single pooled control sample periodically throughout the entire run (once every 20 samples) to account for potential drift in instrument performance (53). Finally, prior to data analysis it is essential to run quality control checks on the data, including assessment of batch effects using multi-dimensional scaling approaches such as principal component analysis (PCA) or uniform manifold approximation and projection (UMAP). If batch effects are present, implementation of correction tools to minimise their effects on interpretation should be considered (53, 54). In some cases where batch effects are too significant to be adequately corrected, it may be necessary to consider re-running samples with appropriate controls.

8. I agree with the authors that the DO population is a valuable resource for mouse genetic studies. However, other mouse populations, such as CCs, BXD, UM-HET3, and HMDP, are also important. These mouse populations can also bridge the gap between studies using the C57BL/6J strain. Therefore, it should not be concluded

that the DO population is the best one, it is one of the useful ones. Additionally, reference 12 does not make a statement that the DO population should be a routine part of the clinical trials pipeline.

References to additional models are now included throughout the manuscript and phrasing relating to DO populations has been toned down, as requested. For example:

- Line 83 - “Largescale genetic and phenotypic analysis of metabolic traits in mice has been popularised through the development of large panels of inbred mice like the BXD (21, 22), UM-HET3 (23), Hybrid Mouse Diversity Panel (HMDP) (24, 25) and Collaborative Cross (CC) (26).”
- Line 324 - “Importantly, several other populations, including the BXD, HDMP (9) and Swiss Outbred (69) resources, contributed greatly to the fundamental characterisation of complex traits.”

We have also amended the reference on line 333-335. Thank you for pointing this out.

9. Line 301: This statement should be toned down, as other populations contributed to fundamental characterizations of complex traits: BXDs, HMDP, Swiss Outbreds, CCs, etc ...

Please see response to comment 8.

10. Lines 318-319: It is not clear why the authors choose those specific examples. Also, in this section a mention to potential genetic difference between mouse and humans for the targets to test is missing. This might not necessarily represent a problem but it can be something to consider in particular for some genes. For example, there are genes identified as good targets in mouse with no orthologs, or with more than one ortholog in humans.

These examples have been removed.

11. Lines 331-333: As stated now, it looks like tools to experimentally validate hypothesis in the field of metabolism are missing although many options exist. A reference supporting this statement is required or the authors should expand more with a set of specific examples compared to what exist for example in the field of neurosciences as reported in the text.

Validation of correlations observed in multi strain studies are complicated by the over-reliance of C57BL/6J mice. While tools such as knockout mouse models, AAVs, ASO, LNP exist, they have largely been optimised for C57BL/6J mice. This is particularly true for knock-out or knock-in models. To make this point clearer we have amended the text. This now reads:

- Although many validation approaches exist, including knockout and knock-in mouse models, adenoassociated viruses (AAVs), lipid nanoparticles (LNPs) and anti-sense oligonucleotides (ASOs), these have been predominantly optimised using C57BL6/J mice and it is unclear how simple it will be to apply such tools to other strains.

While knockout mice are a workhorse of metabolic research, results from such experiments are often not physiologically relevant in the context of natural genetic variation in protein abundance. Additionally, the time constraints of generating knockout mice in different strains or performing other gain/loss of function studies is not compatible with the high throughput nature of target generation that has been facilitated by systems genetics.

A softer tone has also been adopted. For example:

- ... some strains may be protected from a particular metabolic defect because they overexpress a particular protein, while others may be susceptible because they are low expressers. The most powerful validation of this would be to knock-down expression in the protected strain in a tissue-specific manner and overexpress in the latter, however, such experiments are challenging to perform at present.

12. Line 347: This statement is needs to be reformulated. Widely used techniques exist to perform LOF/GOF studies or introduce specific mutations in strains with different genetic backgrounds exist.

Please see response to comment 11. We appreciate that these tools exist but their application to genetically diverse populations is less straight-forward than their application to common lab strains. We have included an example from our own laboratory:

- Further work is required to determine how effectively AAV and LNP approaches will work across different genetic backgrounds, as unpublished data from our lab indicates that the efficiency of AAV transduction varies by mouse strain.

Minor comments:

1. Line 43-44: it could be useful to cite some papers contrasting the examples cited in lines 40-42 but in other strains other than C57BL/6J.

The wording of this sentence was not clear and so it has been restructured:

- Had we chosen something other than the C57BL/6J mouse as our primary model, the field of metabolism would look very different. A/J mice have shown us that obesity does not always go hand in hand with glucose intolerance (4, 5), BALB/C mice have revealed the tissue specific nature of insulin resistance (6), and PWK mice have demonstrated that mice do develop diet-induced NASH akin to humans (7).

2. Line 73: it might be worth to also point out the very large differences naturally existing between mouse subspecies and with more wild-derived strains, such as CAST as also pretty clear from Fig. 1.

This is a good point. The text now reads:

- While studies like this highlight marked differences between mouse subspecies and wild-derived strains compared to their traditional laboratory counterparts, one need not climb so far on the phylogenetic tree to find striking phenotypic variation (19)

3. Line 80: only mentioning genetic analyses as justification for large populations might be reductive as this also concerns for example association studies between molecular traits and phenotypes (such as ePheWAS) and so on.

This was the intended message and we have rephrased. The section introducing DO mice has been renamed “Beyond genetic mapping” and now reads:

- An ever-growing tool kit of genetically diverse mouse populations has enabled high-resolution genetic mapping of complex traits and molecular phenotypes, such as protein and metabolite abundance (31, 55, 59-67). However, genetic mapping is only one of many powerful uses for these populations. We direct interested readers to the following reviews to learn more about the development and genetic architecture of the most recently developed mouse genetic resource, the Diversity Outbred (DO) population (29, 30, 68). Several other populations, including the BXD, HDMP (9) and Swiss Outbred (69) resources, contributed greatly to the fundamental characterisation of complex traits through approaches such as phenome wide association studies.

4. Line 85-86, more representative papers after each mouse population should be added.

Additional references included.

5. Line 93: As a minor comment, as the transition is not formulated from the sentence before it implies that we are easily able to just identify those particular strains. But this assumes we have access to prior data from a set of strains that can vary in size according to the needs. This is partially addressed later (line 129) by mentioning that some large databases of phenotypes exist.

Agreed. The sentence now reads:

- ... many studies including broad explorations of metabolism across genetic backgrounds, can in fact be accomplished without using such large and expensive mouse resources. This hinges on co-operation and the public release to facilitate selection of the most appropriate strains. Fortunately, a number of online resources already exist and should be utilised by metabolism researchers new to the field of mouse genetics (Box 1).

6. Line 139: it might not be clear for some readers what is the benefit for the community if it is not specified that this comes through the public release of the generated data.

We have now specified that: “A broad range of metabolic differences can readily be observed using only a few fixed strains. Utilising these strains and resources, therefore, has advantages for both the user and the community through public release of new data that will inevitably improve the design and outcomes of future experiments.”

7. Line 155: this resource include also studies conducted on other set of strains other than the BXDs.

Please see response to major comment 2.

8. Line 230: this applies in particular to MS even though this section also concerns DNA-sequencing (including RNA-seq if understood correctly). Also, since samples preparation is chronologically before data generation, we recommend to invert the order.

Please see response to major comment 7.

9. Line 216, should "-omics" be "multi-omics"?

Agreed.

10. Line 221, <https://www.science.org/doi/10.1126/science.aad0189>, this paper is also an example for proteomic study in multi-strains

This is a worthwhile addition.

11. Line 242: As a minor comment, for non-MS expert it might be useful to explain what being qualified as zero means.

The rationale has been expanded in-text to: "Most reference libraries are generated using the C57BL/6J mouse genome and, for TMT-based peptide quantification, this has the potential to confound data analysis as coding variants may lead to the resulting peptide being quantified as zero rather than not detected."

12. Line 271, missing reference.

This consideration stems from our experience developing new methods for use with large numbers of diverse mice, although many of these are currently unpublished. We have updated the manuscript to make this clearer:

- "Scale assays for robustness across many mice, as well as financial constraints. In our experience, the most worthwhile optimisations relate to performing assays in 96-well plates and evaluating how the impact of the increased time required to perform assays over large sample sizes may influence results."

13. Line 317: As a minor comment, this is repeated from Line 309

This has been removed.

Referee #2:

General comments

Given the absolute requirement for translatability of mouse data into human relevance in drug development, the choice of proper model systems remains a critical issue, particularly in the field of complex diseases including obesity, diabetes and related metabolic dysfunction. In this respect, the authors address an interesting

and relevant topic in biomedical research. The manuscript is well-structured, concise, and easy-to-read. The figure provides a first and helpful guidance to potentially relevant mouse models in this field. As some aspects in the second half of the manuscript ("challenges of working across strains", "validation bottlenecks") appear somewhat trivial in some parts, the manuscript would strongly benefit from the inclusion of more concrete and successful examples of DO animals in translational research ("In which case did the use of DO animals really make a difference in drug development?"). The addition of specific examples will underline the author's main arguments and make a more convincing case in general.

Thank you for reviewing this manuscript and for seeing the value of including diverse models in metabolism research. While some sections are indeed trivial, they represent critical steps in the transition from classic studies in a single strain to multi strain projects. The importance of these considerations was not immediately apparent to us when we first ventured into multi strain studies and so we thought providing a practical perspective might greatly assist those with inexperienced with multi strain studies. Of course, this may seem common sense to experienced investigators, however, we must encourage as many people as possible to improve the impact of their animal studies if we as a community are to overcome external biases relating to the non-translatability of animal studies and inefficiency of fundamental research more generally.

Unfortunately, there are few examples of the utility of DO mice in drug discovery pipelines. This was precisely the motivation for the "Beyond genetic mapping" section of the commentary as the potential benefits to translation and understanding responders vs non-responders are very clear. Thus, as others have done, we feel it appropriate to advocate for the future use of DO mice in the therapeutic pipeline.

Dear David, dear Harry and Stewart,

Thank you for submitting your amended commentary manuscript for consideration by the EMBO Journal. We have now assessed it in detail and concluded we can swiftly proceed with acceptance and production of the commentary.

I now still need you to address a number of minor formatting issues as detailed below.

Please let me know if you have questions related.

We are looking forward to your final resubmission.

Best regards,
Daniel

Daniel Klimmeck, PhD
Senior Editor
The EMBO Journal

Formatting changes required for EMBOJ-2024-118176R:

>> Authors: corresponding author email address should be added to title page.

>> Figure: please remove the figure from the manuscript and upload as a separate file. Legend should be moved to the end of the manuscript.

>> References: please correct to EMBO Journal style, alphabetical and 10 et al.

Further information is available in our Guide For Authors: <https://www.embopress.org/page/journal/14602075/>

authorguide

The authors addressed the minor editorial issues.

Dear David,

Thank you for sending us the updated final version of the commentary article.

I am pleased to inform you that your manuscript has been accepted for publication in the EMBO Journal.

Further, I will now contact our graphics illustrator to convert the commentary figure into journal style. He will contact you shortly on the proof stage image for your input.

If you have any questions, please do not hesitate to contact me.

Thank you again for your kind contribution to The EMBO Journal, which is much appreciated.

with

Best regards,

Daniel

Daniel Klimmeck, PhD
Senior Editor
The EMBO Journal
